# Physical Activity and Quality of Life in High School Students: Proposals for Improving the Self-Concept in Physical Education

**DOI:** 10.3390/ijerph18137185

**Published:** 2021-07-05

**Authors:** Mikel Vaquero-Solís, Miguel Angel Tapia-Serrano, David Hortigüela-Alcalá, Manuel Jacob Sierra-Díaz, Pedro Antonio Sánchez-Miguel

**Affiliations:** 1Department of Didactics of Musical, Plastic and Body Expression, Faculty of Teaching Training, University of Extremadura, Avenida Universidad, S/N, 10071 Cáceres, Spain; mivaquero89@gmail.com; 2Department of Specific Didactics, Faculty of Education, University of Burgos, CalleVilladiego 1, 09001 Burgos, Spain; dhortiguela@ubu.es; 3Physical Education Department, Faculty of Education, University of Castilla-La Mancha, Campus Universitario, S/N, 16071 Cuenca, Spain; jacobsierradiaz@hotmail.com

**Keywords:** adolescents, physical activity, predictive model, quality of life, students

## Abstract

Adolescence is a critical period for the acquisition of health-related behaviors that will transcend later psychological well-being in adulthood. The present study presents a theoretical model whose objective is to analyze how physical activity predicts an adequate quality of life through self-concept and subjective happiness among adolescents. A total of 452 students aged 12 to 15 (M = 13.8; SD = 0.77) from four Compulsory Secondary Education institutes of the Autonomous Community of Extremadura participated, including boys (*n* = 258) and girls (*n* = 194). The students reported information on the following variables: physical activity, body mass index, self-concept, subjective happiness, and quality of life. The results show acceptable fit indices for the proposed theoretical model, which showed the importance of physical activity through self-concept and subjective happiness in quality of life: MRLχ^2^ = 67.533, *p* < 0.05, CFI = 0.93, TLI = 0.90, SRMR = 0.05, and RMSA = 0.07. Likewise, the model presented a better fit index for males than females. This study draws conclusions on the importance of physical activity as a predictor of quality of life mediated by the perception of self-concept and mood in adolescents.

## 1. Introduction

Adolescence is characterized as a critical stage in the development of health-related habits, the importance of which transcends beyond this period, since they predict health-related behaviors in adulthood [1,2]. This stage is marked by a decrease in levels of physical activity [3], while increasing the time dedicated to sedentary activities [4], causing, among other consequences, an increase in overweight and obesity in children and adolescents [5]. In addition, it has been shown that the increase in sedentary time not only affects physical factors, but also affects the correct development of psychological well-being [6]. In this regard, physical activity carried out in the school context plays an important role in the physical and psychological well-being of children and adolescents [7], contributing to both the improvement of body composition and physical condition [8], such as improving those constructs related to psychosocial well-being [9]; satisfaction with life [10]; mental well-being [11]; and the self-concept [12]. In this sense, it is important to highlight the role of the teacher and learning styles as an important figure in the development and promotion of healthy lifestyle [13]. Specifically, styles such as gamification [14] or cooperative learning [15] have focused on promoting healthy habits to improve body composition, physical activity, self-concept, and quality of life.

The concept self-concept is a multidimensional construct that contains attitudes and feelings about the capacities, as well as appearance and social acceptability of individuals [16]. It can be defined as the perception that individuals have of their abilities and physical appearance [17], and it is one of the most relevant vital constructs for psychological well-being during adolescence [18,19]. In this regard, previous studies have shown the strong relationship between physical activity and body mass index (BMI) with self-concept [20,21,22,23]. However, it is important to mention the role that physical activity plays in the perception of an adequate self-concept [20]. Longitudinal research based on the promotion of physical activity and carried out in the school context to improve self-concept has verified these associations [21,23,24]. The intervention program developed by Rey et al. [21] in adolescents showed differences in the self-concept in favor of those participants who had carried out vigorous physical activity. Similarly, previous investigations [24,25,26] showed that the application of evaluation of intervention programs focused on promoting physical activity showed an improvement in self-concept in children and adolescents.

Likewise, it is important to highlight the mediating role of self-concept, acting in some cases as a mediator between motor skills and physical activity [23,27], and in other cases acting as a mediator between physical activity and psychological variables [28]. In the same way, it is worth highlighting self-concept as a key construct to improve well-being during adolescence [29], presenting a close relationship with the moods and emotions that influence the subjective well-being of adolescents [30]. In this regard, a correct regulation of emotions in the development of self-concept during the school stage allows for maintaining a positive self-concept in different areas: academic, social and physical [31,32]. For this reason, previous research highlighted the incidence of self-concept as a predictor of subjective happiness, specifically the dimensions of self-esteem, physical appearance and self-confidence [18,33,34,35].

On the other hand, the positive emotion of subjective happiness refers to a general evaluation of well-being [36]. In this regard, it is important to highlight the impact it has on domains related to general health, such as subjective well-being, life satisfaction, and health-related quality of life [37]. Some authors propose satisfaction with life and quality of life as similar interdependent concepts, exposing the fact that satisfaction with life is a global assessment of quality of life based on the criteria established by the individual [38]. In this sense, the study carried out by Abdel-Khalek [39] showed that high levels of subjective happiness and life satisfaction were related to a better quality of life. Finally, it is important to point out the mediating and predictive value of subjective happiness in health-related quality of life [40].

According to previous studies, there is little research that has jointly addressed such important aspects of psychosocial well-being (quality of life, subjective happiness and self-concept) in adolescents and the performance of physical activity, since in some studies they are postulated as similar indicators of psychological well-being, and in other studies postulate it as different subdomains that make up well-being valued through quality of life or similar variables. In addition, previous research has highlighted the mediating value of self-concept [20,41] and subjective happiness [34,42,43]. In this sense, it is important to highlight the positive impact that physical activity has on the mental health of adolescents. In this regard, some authors emphasize the intensity of physical activity practice to improve self-concept [23]; however, other authors who do not take self-concept into account highlight the importance of physical activity in mental health [44]. Similarly, few studies have evaluated the associations that the perception of self-concept has in quality of life [45,46], and those that exist address these associations from the point of view of certain subdomains of self-concept such as self-esteem, which has been positively related to quality of life [47,48]. Likewise, Gonzalo-Silvestre’s study [45] also showed self-concept as a good predictor of quality of life. Therefore, the present research aims to contribute to the improvement in mental well-being in children and adolescents through the proposal of a theoretical model based on the importance of physical activity as a predictor of health-related quality of life [49] through constructs that affect the development of the psychological well-being of adolescents [50], such as subjective happiness and quality of life.

### Theorical Model Proposal

The present proposed theoretical model was based on scientific evidence for its elaboration. In this regard, the following model postulates the role of physical activity through self-concept and subjective happiness in the quality of life of children and adolescents. To justify the model, we relied on the relationship and predictive values of some variables with others. Likewise, previously published theoretical models have served as justification in the establishment of order of the variables [20,45,50]. For all these reasons, the present model proposes that physical activity and body mass index are related to and can predict self-concept, subjective happiness, and quality of life. Along these lines, several investigations show the importance of physical activity in the well-being of children and adolescents [51,52,53]. However, the benefits of this are due both to the time spent practicing physical activity and its intensity [20,21]. Both ways of considering physical activity influence the perception of self-concept, making it better or worse [29], which, according to previous studies, could have effects on the well-being and emotional state of children and adolescents [18,34,35] that would explain the quality of life they perceive [49].

Therefore, the following research question arises: can physical activity and body mass index explain the quality of life of adolescents through the mediating factors of self-concept and subjective happiness? In this sense, the objective of this research is to analyze the explanatory value of physical activity on quality of life through self-concept and subjective happiness. In addition, it is proposed to determine the validity of the theoretical model for both sexes. In this regard, it is thought that physical activity and body mass index will predict quality of life through the mediating factors of self-concept and subjective happiness.

## 2. Materials and Methods

### 2.1. Design and Participants

The present study presents a predictive cross-sectional design, since it aims to establish associations and predictions in the study variables. The sample was comprised of 452 students from four Compulsory Secondary Education (ESO) institutes in Extremadura (Spain), aged between 12 and 15 years (M = 13.87; SD = 0.77), of which 57.08% were boys (*n* = 258) and 42.92% were girls (*n* = 194). The selection of the sample was carried out through intentional sampling for convenience according to the distance of the schools to the research staff in charge of data collection, the willingness to collaborate on the part of the teaching staff, and the time required for the researcher to travel towards collaborating centers. Likewise, all subjects consented to their participation in the study.

### 2.2. Measures

Physical activity. Physical activity was measured using the International Physical Activity Questionnaire–Short Form (IPAQ-SF). This questionnaire assesses physical activity in the last 7 days [54]. In this sense, the evaluation of the score was carried out through the calculation of the scores obtained according to the minutes of physical activity practiced per day by each participant. To obtain this score, the calculation was obtained based on the international guide for processing and IPAQ data analysis (International Physical Activity Questionnaire. Available in: https://sites.google.com/site/theipaq/scoring-protocol) (accessed on 7 June 2019), in which different scales and ranking are established according to the METS (metabolic index measure) for each type of intensity in the physical activity carried out.

Body mass index. Body weight and height were assessed when students were barefoot and wearing underclothes. Weight was recorded to the nearest 0.1 kg using an electronic scale (model SECA 877), and height was assessed to the nearest 1 mm using a telescopic height-measuring instrument (model SECA 217). Both measurements were performed twice, and averages were used.

Physical self-concept. Physical self-concept was measured with the Spanish version of the Physical Self-Perception Profile [55,56]. This instrument comprises 28 items that assess five factors: fitness (five items, e.g., “I feel very confident to practice continuously and to maintain my physical shape” α = 0.78); perceived competence (four items, e.g., “I am very good at almost all sports” α = 0.78), physical strength (six items, e.g., “when it comes to situations that require strength, I am the first to offer myself” α = 0.68), appearance (nine items, e.g., “I feel very satisfied with how I am physically” α = 0.72), and self-esteem (four items, e.g., “when it comes to physical appearance I do not I feel very confident in myself ” α = 0.65). Finally, the instrument showed a total alpha of α = 0.89 for the present study.

Subjective happiness. The subjective happiness was obtained with the Spanish Version of the Subjective Happiness Scale (SHS) [57]. This scale is composed of 4 items that assess subjective happiness in a global way through statements with which the participants rate themselves and compare themselves with others. All items are valued using a 7-point Likert scale, where 1 (totally disagree) and 7 (totally agree). The questionnaire showed adequate reliability α = 0.82.

Quality of life related to health. The quality of life was assessed through the KIDSCREEN-10 questionnaire [58]. The Kidscreen-10 is a one-dimensional version and represents an overall quality of life score. Items 1 and 2 explore the participant’s level of physical activity and fitness, respectively. Items 3 and 4 reports on the absence of feelings such as loneliness and sadness. Items 5 and 6 refer to one’s own freedom in relation to the age to choose. Item 7 explores the relationship between the child and parental figures. Item 8 explores the quality of interaction between the child and their classmates. Finally, items 9 and 10 explore the child’s perceptions about their cognitive ability, learning and concentration. All items were evaluated through a 5-point Likert scale, and reported a reliability of α = 0.79.

### 2.3. Procedure

The present study was carried out in several phases. In the first place, the educational centers were contacted to inform them about the aim of study and determine their involvement in it. Once permission was obtained from the educational center, informed consent was provided to the students so that the parents of the participants could complete it, authorizing their respective children to participate in the project. Finally, the questionnaires were administered to the group of students so that they completed them while the researcher explained the purpose of the project to the students and remained in the classroom in case any questions arose. The questionnaire was completed in approximately 25 min. Likewise, the research was carried out in accordance with the Declaration of Helsinki, and the study was approved by the Ethics Committee of the University of Extremadura (145/2019).

### 2.4. Data Analysis

For data analysis, the statistical package SPSS Version 0.23 (SPSS Inc., Chicago, IL, USA) was used, in which descriptive statistics and normality tests were previously carried out. In this sense, the Kolmogorov–Smirnov test, the Rachas random test, and the homoscedasticity or Levene’s test for equality of variances were performed. The results of the normality tests recommended the use of parametric tests. Likewise, the association between the study variables was examined using bivariate correlations (Pearson).

The statistical package Mplus 7.0 (MuthenyMuthen, 2012) was used to verify the predictive capacity of physical activity on self-concept, subjective happiness, and quality of life, through the elaboration of a structural equation model. Subsequently, the predictive capacity of the model for both genders was verified.

Finally, to test the indirect effects, the model was re-estimated using bootstrapping resampling procedures (*n* = 5000) in order to find the 95% bias correction confidence intervals (BcCI 95%) [59]. If the 95% BcCI did not include zero, the indirect association was considered significant. This model was estimated using the maximum likelihood estimator (ML), as bootstrapped MLR is not yet available in Mplus.

## 3. Results

Table 1 shows the descriptive statistics of all the variables in relation to gender. In this sense, higher values were shown in each of the variables in favor of the male gender. Likewise, the correlations show negative associations between BMI and the dimensions of physical appearance and physical condition (all, *p* < 0.01) of self-concept. Similarly, it was also positively associated with the perceived strength of the self-concept dimension (*p* < 0.01). On the other hand, physical activity was positively related to all dimensions of self-concept, quality of life and subjective happiness (all, *p* < 0.05).

### Structural Equation Model

Below, a model of structural equations is presented (Figure 1) based on the theoretical postulates presented in the introduction with which it is intended to justify its elaboration. This model postulates physical activity and BMI as independent variables that predict the quality of life of adolescents through the perception of their self-concept and mood.

The initial model (Figure 1) was developed through the creation of four variables and a latent variable (physical self-concept) formed by physical appearance, physical condition, perceived competence, self-esteem, and perceived strength. This initial model showed the following fit indices, MRLχ^2^ = 187.099, *p* < 0.05, CFI = 0.90, TLI = 0.87, SRMR = 0.09, and RMSA = 0.10, which were not acceptable; therefore, it was decided to restructure the model following the principles of previous research where such predictions are justified, such as the following: the predictive role of perceived competence in self-esteem [60]. In this sense, the model was restructured based on the proposals of the theoretical model presented by Fox and Corbin (1989) [56], establishing self-concept as a multidimensional variable formed by different domains, whose most important domain is self-esteem followed by perceived competence, fitness, perceived strength, and appearance. In this sense, perceived competence is the extent to which one judges oneself as capable in a certain area of life, which is why the higher the perceived competence, the greater the probability that they will present good self-esteem [60]. Consecutively, it was adjusted according to the effect of physical condition on quality of life in addition to its role in mental well-being [51]. Finally, the importance of BMI in appearance was assessed [61,62,63]. In this regard, the final model (Figure 2) showed acceptable fit indices: MRLχ^2^ = 67.533, *p* < 0.05, CFI = 0.93, TLI = 0.90, SRMR = 0.05, and RMSA = 0.07, showing the effect of a higher incidence of physical activity on quality of life.

The validity of the model was checked for each gender in order to assess the invariance of the model. The fit indices showed that the model was acceptable for the boys: MRLχ^2^ = 45.716, *p* < 0.05, CFI = 0.94, TLI = 0.90, SRMR = 0.05, and RMSA = 0.07, but not for the girls: MRLχ^2^ = 49.638, *p* < 0.05, CFI = 0.91, TLI = 0.86, SRMR = 0.06, and RMSA = 0.09.

Indirect effects between variables showed that BMI was negatively and non-significantly related to quality of life (*p* = 0.08) through self-concept and subjective happiness (β = −0.052, 95% BcCI = −0.102, −0.003), while physical activity was significantly positively related (*p* = 0.04) to quality of life through self-concept and subjective happiness (β = 0.062, 95% BcCI = 0.027, 0.097). Subsequently, the BMI was negatively but not significantly (*p* > 0.10) related to subjective happiness through self-concept (β = −0.042. 95% BcCI = −0.085, 0.001), while physical activity was significantly positively related to it (*p* < 0.00) (β = 0.050, 95% BcCI = 0.019, 0.080). Finally, self-concept was related to quality of life through subjective happiness in a significant positive way (*p* > 0.00). (β = 0.390, 95% BcCI = 0.325, 0.454).

## 4. Discussion

The present study aimed to examine the predictive value of physical activity and BMI on health-related quality of life through the perception of self-concept and subjective happiness. The main findings show the validity of a theoretical model with acceptable fit indices, in which a higher amount of physical activity and BMI through self-concept and subjective happiness predicted health-related quality of life in adolescents.

In line with the results, our hypothesis proposed that physical activity would predict health-related quality of life through the mediating factors of self-concept and subjective happiness. In this sense, our study showed that physical activity was significantly associated with quality of life. The results of previous studies are in line with our findings, emphasizing the importance of the intensity of physical activity for it to present a significant effect on quality of life [49,64,65]. A possible explanation for this fact is that the realization of physical activity supposes a positive impact on the state of mind, which directly affects the perception that people have of their health-related quality of life. Likewise, we must highlight the proximity between the terms quality of life and satisfaction with life [38,43]. In this sense, in line with our results, [50] pointed out that vigorous moderate physical activity affects well-being and states that this effect is mediated by other variables such as self-esteem, self-efficacy and the social support received.

Additionally, the results found highlight the positive relationships established between physical activity, self-concept and subjective happiness. These results are consistent with those found in previous studies where physical activity was significantly positively related to subjective happiness and self-concept [66,67]. Similarly, the results of the present study also show a positive association between self-concept and subjective happiness. A possible explanation for these results may be the fact that the practice of moderate–vigorous physical activity plays a fundamental role in improving self-concept, promoting a better perception of their body image, which could be accompanied by an improvement in the factors that make up self-concept. In this regard, the meta-analysis carried out by Rodriguez-Ayllón [52] highlights the role of physical activity on mental well-being in children and adolescents, associating physical activity with self-image, satisfaction with life and happiness. Likewise, several studies have shown the benefits of physical activity in improving moods. In this regard, several biological mechanisms could explain the effect of physical activity on mood [68]. Physical activity can improve monoamine levels by increasing endorphin levels [69] and decrease cortisol secretion levels [70]. Likewise, it is important to take into account active methodologies that make students a protagonist in the acquisition of healthy habits through positive experiences that amuse them while consolidating the healthy habits learned [14].

Regarding the mediating value of self-concept and subjective happiness in the quality of life, the results of the indirect effects of self-concept and subjective happiness on the quality of life are significant. These results are consistent with those found in Fernández-Bustos et al. [20] and Jekauc et al. [27], where the mediating value of self-concept in relation to physical activity was confirmed. Likewise, previous studies are consistent with our results, highlighting the importance of self-concept and subjective happiness in quality of life [39,41,43]. In this regard, efforts to seek development in the feeling of happiness have an effect on satisfaction with life [43]. However, due to the heterogeneity of the sample, the context, or the lack of analyses that indicate predictions of the effect of physical activity on quality of life through subjective happiness, there is not much literature on the matter that supports the hypothesized relationships and allows us to discuss these findings extensively. In this sense, Gonzalo-Silvestre and Ubillos Landa [45] showed how, in a female population, self-concept mediates the relationships between physical activity and quality of life, distinguishing itself from subjective well-being or satisfaction with life. In this regard, the presence of positive psychological dimensions such as subjective happiness or life satisfaction is related to the general self-concept [40].

Finally, it is important to note that this model presented fit indices that are more acceptable for boys than for girls. In this regard, there are several causes that could explain these findings, such as adolescence, which is characterized by a decrease in levels of physical activity, especially among girls [71,72]. This fact could be explained through the amount of daily physical activity performed by boys or girls [73], or the different types of physical activity practiced according to gender, which is associated with gender stereotypes, causing the rejection of certain activities due them being perceived as masculine [74]. In this regard, girls tend to prefer activities related to body shape and health with a more aesthetic orientation, preferring individual sports, while boys tend to opt for activities focused on improving fitness or physical performance, choosing team sports in which strength and competitiveness predominate [75,76]. In the same way, another possible explanation can be found in the perception of weight and body mass index, which affect girls more significantly than boys, causing damage to self-esteem according to their perceived appearance and affecting their well-being [77]. In this regard, the study by Vaquero-Solís et al. [78] based on the theory of self-objectification shows that physical activity in girls is more associated with appearance and self-esteem.

The results of this study show a complex theoretical model which highlights the importance of physical activity in adolescents for obtaining an adequate perception of self-concept and a more positive state of mind [79], which in turn could translate into an improvement in health-related quality of life. These results should be considered with caution, since this study has some limitations such as its cross-sectional nature, meaning that it does not allow us to establish cause-effect relationships, and the instruments for assessing physical activity, which do not allow an objective assessment of the variable. Regarding the strengths, this study has contributed to increasing the existing literature through its contributions on two research topics. On the one hand, it contributes to increasing the scientific knowledge that relates to physical activity, body mass index and self-concept with variables of well-being in the adolescent population (moods, satisfaction with life, subjective well-being) [29,30]. Additionally, on the other hand, it contributes to the small number of studies that relate self-concept to quality of life [45]. In addition, a predictive model is provided in a mixed adolescent population, as a similar model had previously been tested in a population of adult women [45]. Similarly, the indirect effects of physical activity and BMI with quality and life are shown through the partial mediating value of self-concept and mood. Finally, it contributes to providing an explanation of well-being in the male and female gender derived from the types of physical activity and the importance of appearance and self-esteem in one gender and another. Thus, we believe that our model contributes something new to the literature, since its practical application could contribute to the improvement of physical and mental well-being in children and adolescents. In such a way that the improvement of the attributes of physical condition derived from the practice of physical activity would lead to an improvement in psychological health through a better perception of their well-being and quality of life.

## 5. Conclusions

In conclusion, the importance of physical activity as a predictor of quality of life mediated by the perception of self-concept and mood in adolescents is evident. In this respect, our study has shown through a structural equation model the important role that the practice and intensity of physical activity have in the perception of self-concept and state of mind, for a more favorable perception of the quality of life. As practical implications, this theoretical model highlights the importance of performing physical activity with an adequate frequency and moderate vigorous intensity from an early age, since this fact contributes to adequate mental development, improving the quality of life and avoiding, in stages of subsequent development, different types of disorders in relation to their perception of self-concept. Similarly, this theoretical model can contribute to promoting new methodologies when teaching classes in different subjects, incorporating innovative methodologies that favor the acquisition of knowledge through the practice of physical activity. In this way, we would recommend increasing the daily minutes of physical activity and promoting mental well-being derived from physical activity. In the same way, it is important to support interest in the different forms of physical activity for the promotion of sports practice within the educational context, since this would avoid the rejection of more sedentary people and with a worse perception of their self-concept, promoting the performing physical activity, with a comprehensive approach to education for the promotion of well-being in oneself and with others. Future studies should be directed towards the realization of a longitudinal intervention in the Physical Education class that encourages moderate–vigorous physical activity to improve the perception of self-concept and moods in the initial educational stages, with the aim of contributing to a correct development of the person. Likewise, it would be convenient for social agents (teachers, parents and peers) to promote the practice of physical activity, indirectly interceding the problem of overweight and obesity in children and adolescents.

## Figures and Tables

**Figure 1 ijerph-18-07185-f001:**
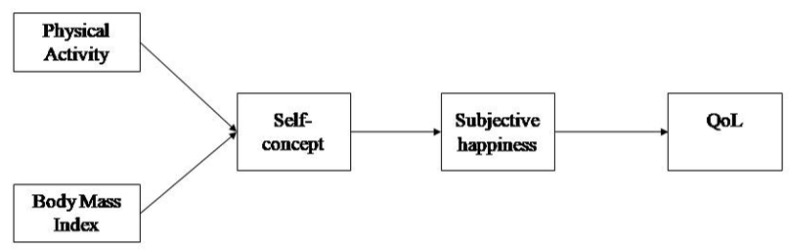
Hypothesized model in adolescents. **Note.** QoL: Quality of life.

**Figure 2 ijerph-18-07185-f002:**
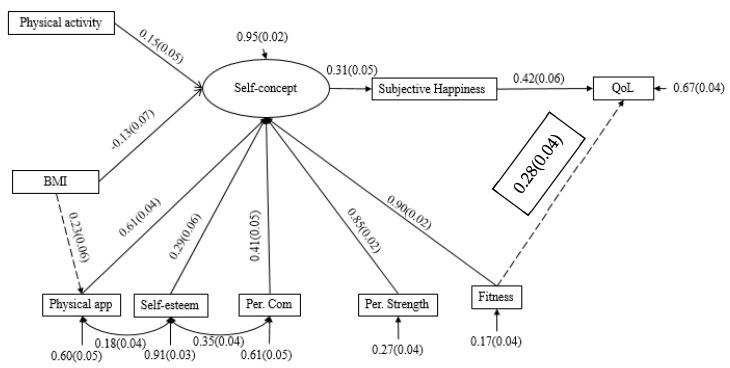
Structural equation model. **Note.** QoL: quality of life; BMI: body mass index, physical app: physical appearance, per.com: perceived competence; per strength: perceived strength.

**Table 1 ijerph-18-07185-t001:** Descriptive statistics and correlation analysis between the study variables.

Variables	Boys *n* = 258	Girls *n* = 194	Total	*p*	Pearson’s Correlation									
	*M ± SD*	*M ± SD*	*M ± SD*		1.	2.	3.	4.	5.	6.	7.	8.	9.	10
1. Body mass index (kg/m^2^)	21.87 ± 4.42	20.93 ± 3.32	21.48 ± 4.02	0.02	-	0.01	−0.07	−0.25 **	0.14 *	−0.07	−0.12 **	−0.10 *	−0.08	−0.06
2. Physical activity	1749.21 ± 0.80	1511.59 ± 0.75	1647.22 ± 0.79	0.01		-	0.33 **	0.20 **	0.26 **	0.13 **	0.38 **	0.36 **	0.10 *	0.20 **
3. Perceived competence (1–10)	6.77 ± 2.16	5.69 ± 2.29	6.31 ± 2.28	0.00			-	0.40 **	0.51 **	0.23 **	0.77 **	0.81 **	0.20 **	0.28 **
4. Physical appearance (1–10)	6.31 ± 1.69	6.01 ± 1.85	6.18 ± 1.76	0.07				-	0.34 **	0.48 **	0.48 **	0.72 **	0.34 **	0.31 **
5. Perceived strength (1–10)	6.29 ± 1.69	5.56 ± 1.56	5.98 ± 1.64	0.00					-	0.31 **	0.48 **	0.69 **	0.11 *	0.14 **
6. Self-esteem (1–10)	6.79 ± 2.08	6.53 ± 2.00	6.68 ± 2.04	0.18						-	0.27 **	85 **	0.09 *	0.14 **
7. Fitness	6.78 ± 1.99	6.11 ± 2.02	6.50 ± 2.03	0.00							-	0.83 **	0.21 **	0.37 **
8. Global self-concept score (1–10)	6.59 ± 1.42	5.98 ± 1.42	6.33 ± 1.45	0.00								-	0.26 **	0.34 **
9. Subjective happiness (1–7)	5.49 ± 1.06	5.29 ± 1.14	5.40 ± 0.1.10	0.04									-	0.47 **
10. Quality of life (1–5)	3.82 ± 0.59	3.70 ± 0.62	3.77 ± 0.61	0.03										-

**Notes:** * *p* < 0.05; ** *p* < 0.01.

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
