# Peer review of "Physical Activity and Quality of Life in High School Students: Proposals for Improving the Self-Concept in Physical Education"

_ijerph, 2021, doi:10.3390/ijerph18137185_

Round 1

Reviewer 1 Report

Physical activity and quality of life in high school students:  proposals for improving the self-concept in Physical Education

General comments

The authors examined the predictive value of physical activity and BMI on health-related quality of life through the perception of self-concept and subjective happiness. Their findings demonstrates the validity of a theoretical model with acceptable fit index, in which a higher amount of physical activity and BMI through self-concept and subjective happiness predicted health-related quality of life in adolescents. This predictive model has contributed to the body of knowledge on physical activity in adolescents and its effects on self-concept, subjective happiness, and quality of life.

Specific, but minor comments

Introduction

Line 28: I suggest you replace “by being” to ‘as’ a critical stage….

Line 30: Instead of repeating characterised, substitute it word ‘marked with’

Line 76: Add the word ‘to’ before the word previous to read: According to previous …

Line 79: The “in others they postulate it” is not clear. Rephrase it

Lines 82-83: “… the associations that the perception of self-concept has in quality of life”. This is not clear. Rephrase it.

Materials and Methods

Line 97: The word “composed” should be ‘comprised’?

Line 119: Something not correct here: [51,52][52]. Again, the word “composed” should be ‘comprised’.

Line 142: I suggest you change the word “object” in the sentence to either ‘objective’ or ‘aim’.

Line 153: Write “V. 23” in full as ‘version’.

Line 164: I suppose “(Preacher and Hayes, 2008)” is a reference. Then, the in-text reference format of the journal is violated here.

Discussion

Line 264: This is not clear “the herself.” Again, the word starting the next sentence in that line: “. “According”. I suggest ‘Regarding the strengths….

Conclusion

Lines 271-273: The sentence is not clear: “It is concluded on the importance of physical activity in adolescents, since this could provide an improvement in the factors that make up self-concept and mood, thus affecting their quality of life.  physical activity in adolescence as a precursor to quality of life through self-concept and subjective happiness.”

Author Response

#Reviewer 1

Dear reviewer, the authors appreciate your assessment and constructive approach to the suggestions for improvement, as they contribute to providing greater quality and robustness to this manuscript. Along the following lines, we hope to provide answers to each of your suggestions for improvement. Therefore, the manuscript has undergone extensive modifications, which have been marked in red instead of with the change control. Finally, thanks for the opportunity to make this article more robust.

General comments

The authors examined the predictive value of physical activity and BMI on health-related quality of life through the perception of self-concept and subjective happiness. Their findings demonstrate the validity of a theoretical model with acceptable fit index, in which a higher amount of physical activity and BMI through self-concept and subjective happiness predicted health-related quality of life in adolescents. This predictive model has contributed to the body of knowledge on physical activity in adolescents and its effects on self-concept, subjective happiness, and quality of life.

Specific, but minor comments

Introduction

Reviewer: Line 28: I suggest you replace “by being” to ‘as’ a critical stage….

Responds: thanks for your suggestion, we have made the changed: pag 1, line 29 “as” a critical stage

Reviewer: Line 30: Instead of repeating characterized, substitute it word ‘marked with’

Responds: Dear reviewer, we appreciate your suggestion for improvement. In this sense we have replaced characterized by “marked with”.Page 29, line 31.

Reviewer: Line 76: Add the word ‘to’ before the word previous to read: According to previous …

Responds: thanks for your suggestion, we have made the changed: pag 2, line 77

Reviewer: Line 79: The “in others they postulate it” is not clear. Rephrase it

Responds: many thanks for your suggestion, we have heeded your suggestion and I think we have fixed the meaning of the phrase by adding the word “studies” after others, pag 2, line 80.

Lines 82-83: “… the associations that the perception of self-concept has in quality of life”. This is not clear. Rephrase it.

Responds: We welcome your suggestion for improvement. However, it is difficult to show more results of the two references that we indicate since we do not have more literature with which to justify these relationships. In this sense, in order to respond to your suggestion for improvement, we have added something else to the phrase that justifies the relationship of self-concept with quality of life, emphasizing the predictive value of self-concept in quality of life.

Pag 2, line 86 “Likewise, Gonzalo-Silvestre's study also showed self-concept as a good predictor of quality of life”

Materials and Methods

Reviewer: Line 97: The word “composed” should be ‘comprised’?

Responds: thanks for your suggestion. In our opinion, we think both meanings are allowed. However, we have substituted compose for comprised. Pag 3, line 116

Reviewer: Line 119: Something not correct here: [51,52][52]. Again, the word “composed” should be ‘comprised’.

Responds: Thanks for your contribution to the authors we have removed [52] and replaced composed with comprised page 3, line 138

Reviewer: Line 142: I suggest you change the word “object” in the sentence to either ‘objective’ or ‘aim’.

Responds: We appreciate your suggestion in this sense we have changed “object” for “aim”: Pag 4, line 161

Reviewer: Line 153: Write “V. 23” in full as ‘version’.

Responds: We appreciate your suggestion for improvement and we have write a version. 23 instead of V. 23. Pag 4, line 172.

Reviewer: Line 164: I suppose “(Preacher and Hayes, 2008)” is a reference. Then, the in-text reference format of the journal is violated here.

Responds: Thank you very much for the review. In this sense, we have added the following reference adapted to the journal format:

Preacher, K.J.; Hayes, A.F. Asymptotic and resampling strategies for assessing and comparing indirect effects in multiple mediator models. Behav. Res. Methods 2008, 40, 879–891

Discussion

Reviewer: Line 264: This is not clear “the herself.” Again, the word starting the next sentence in that line: “. “According”. I suggest ‘Regarding the strengths….

Responds: Thanks for your comment. In this sense we have changed “the herself” for “the variable”, and we have replaced “According” for “Regarding the strengths”. Pag 7, line 288-289.

Conclusion

Reviewer: Lines 271-273: The sentence is not clear: “It is concluded on the importance of physical activity in adolescents, since this could provide an improvement in the factors that make up self-concept and mood, thus affecting their quality of life.  physical activity in adolescence as a precursor to quality of life through self-concept and subjective happiness.”

Responds: We greatly appreciate your suggestion for improvement. The conclusion sentence has been changed, since it was a bit ambiguous and did not respond to the objective of the work. In this regard, the new phrase is the next:

Pag 7, line 315-319 “It concludes on the importance of physical activity as a predictor of quality of life mediated by the perception of self-concept and mood in adolescents. In this respect, our study has shown through a structural equations model the important role that the practice and intensity of physical activity presents in the perception of self-concept and state of mind, for a more favorable perception of the quality of life.”

Reviewer 2 Report

This study deals with a very interesting topic, adding important information on the importance of physical activity on quality of life during a critical period of life: adolescence.

The design of the study is well structured, the methodology and statistical analyses are correct.

In addition, the results and conclusions are clear and exhaustive.

I have only some minor revisions to request.

Abstract: please add the information regarding the provenience of the sample and number of females and males.

Introduction: please check the punctuation; some spaces are missing (lines 30,31,36, 38, 52…)

Please add the following citations:

Bermejo-Cantarero A, Álvarez-Bueno C, Martínez-Vizcaino V, Redondo-Tébar A, Pozuelo-Carrascosa DP, Sánchez-López M. Relationship between both cardiorespiratory and muscular fitness and health-related quality of life in children and adolescents: a systematic review and meta-analysis of observational studies. Health Qual Life Outcomes. 2021 Apr 21;19(1):127. doi: 10.1186/s12955-021-01766-0.

Hayward J, et al. When ignorance is bliss: weight perception, body mass index and quality of life in adolescents. Int J Obes (Lond). 2014. PMID: 24824556 Free PMC article.

Measures: lines 118-139. The Authors reported the alpha di Cronbach for many factors. Was the test of reliability carried out by the Authors or by previous Authors. I think it is necessary to specify it for the readers.

Author Response

#Reviewer 2

Authors: Dear reviewer, the authors appreciate your assessment and constructive approach to the suggestions for improvement, as they contribute to providing greater quality and robustness to this manuscript. Along the following lines, we hope to provide answers to each of your suggestions for improvement.

Reviewer: This study deals with a very interesting topic, adding important information on the importance of physical activity on quality of life during a critical period of life: adolescence.

The design of the study is well structured, the methodology and statistical analyses are correct.

In addition, the results and conclusions are clear and exhaustive.

I have only some minor revisions to request.

Reviewer: Abstract: please add the information regarding the provenience of the sample and number of females and males.

Response: Thanks for your contribution. In this regard we have modified the next sentence in the abstract, we have added the provenience of the sample and number of boys and girls:

Abstract line 17-19: A total of 452 students aged 12 to 15 (M = 13.8; SD = 0.77) from four Compulsory Secondary Education institute of the Autonomous Community of Extremadura participated, boys (n = 258) and girls (n = 194).

Reviewer: Introduction: please check the punctuation; some spaces are missing (lines 30,31,36, 38, 52…)

Response: we appreciate your suggestion of improve and we have reviewed the punctuation and spaces.

Reviewer: Please add the following citations:

Bermejo-Cantarero A, Álvarez-Bueno C, Martínez-Vizcaino V, Redondo-Tébar A, Pozuelo-Carrascosa DP, Sánchez-López M. Relationship between both cardiorespiratory and muscular fitness and health-related quality of life in children and adolescents: a systematic review and meta-analysis of observational studies. Health Qual Life Outcomes. 2021 Apr 21;19(1):127. doi: 10.1186/s12955-021-01766-0.

Hayward J, et al. When ignorance is bliss: weight perception, body mass index and quality of life in adolescents. Int J Obes (Lond). 2014. PMID: 24824556 Free PMC article.

Response: Thanks for your contribution, we have added the following references in the method and results sections: Line 102, and 219

Reviewer: Measures: lines 118-139. The Authors reported the alpha di Cronbach for many factors. Was the test of reliability carried out by the Authors or by previous Authors. I think it is necessary to specify it for the readers.

Response: Thank you very much for your suggestion and we have clarified the phrase in the text explaining that the Alpha is from the present study.

Reviewer 3 Report

  1. Abstract

The research findings are not presented in the abstract. Please report the results in accordance with the research purpose.

  1. The motivation for study is weak. Just because previous research has not examined the relationship that the authors did in the study, does not mean that the study should be conducted. The authors should provide more convincing reasons as to why such relationships should be examined and how and why the results contribute to the literature and practice.
  2. The theoretical background is underdeveloped. The authors should create a theoretical background section and provide in-depth discussions on each construct in terms of how the constructs have been understood and examined in previous studies. Also, it’s informative to talk about the importance of physical activity in high school students.
  3. Hypotheses developments are not provided. Please create another section and provide theories and literature that can support the hypotheses. Please also discuss why physical activity and BMI do not directly influence QOL or subjective happiness. Similarly, why did self-concept not directly affect QOL?
  4. I’m afraid to say that the findings look obvious, and thus I don’t see much contribution of the paper to the literature and practice. I believe there are many articles (as the authors cited) found similar results, and I question whether this paper can make a meaningful contribution that other studies have not discovered. Please discuss how and why the paper can advance our understanding in the area.
  5. The authors said MRLx2 = 187.099, p< .05, CFI = .90, TLI = .87, SRMR = .09, and RMSA = .10 were not acceptable. Based on what evidence did the authors make the decision? There is no definitive fit index, meaning that it is just a mere recommendation. I don’t see any issues with the results, and I believe the authors should proceed with the original model.
  6. “the predictive role of perceived competence in self-esteem [55], physical condition on quality of life and finally the importance of BMI on appearance [56].” This is not good at all. Such a decision has to be based on a priori theory, not second through. I don’t understand how sub-constructs of self-concept can influence each other. Why did only BMI was expected to influence physical appearance? This is ill-theorized and not acceptable. Also, 5 sub-constructs of self-concept were conceptualized as formative (allows towards self-concept). Please provide rationales on the conceptualization.
  7. No practical implications were provided. Please discuss how the findings can help school and governmental policies.

Author Response

Dear reviewer, we welcome your suggestions to improve the quality of the manuscript. We hope we have listened to all his suggestions. In this sense, all the questions have been answered and argued. Therefore, the manuscript has undergone extensive modifications, which have been marked in red instead of with the change control. Finally, thanks for the opportunity to make this article more robust.

Reviewer #3

  1. Abstract

Reviewer: The research findings are not presented in the abstract. Please report the results in accordance with the research purpose.

Response: We appreciate your suggestions. In this sense we have added the following sentence, which predicts the results in the abstract by showing the fit index of the model:

Abstract, line 21 “which showed the importance of physical activity through self-concept and subjective happiness in quality of life: MRLx2 = 67.533, p <.05, CFI = .93, TLI = .90, SRMR = .05, and RMSA = .07.”

  1. Introduction

Reviewer: The motivation for study is weak. Just because previous research has not examined the relationship that the authors did in the study, does not mean that the study should be conducted. The authors should provide more convincing reasons as to why such relationships should be examined and how and why the results contribute to the literature and practice.

Response: the authors appreciate your contribution of improvement. In this sense, in addition to the few investigations that have jointly addressed the variables of our study, it is important to highlight the impact of physical activity on the well-being and mental health of adolescents (Ebdolls et al., 2018) (Justification in the manuscript line text 77 to 81). In this regard, some authors emphasize the intensity of physical activity practice to improve self-concept (Gran et al., 2019), however, other authors who do not take self-concept into account highlight the importance of physical activity in mental health (Hale et al., 2021). Therefore, according to previous research, we believe that it is important to create a theoretical model that considers most of the aspects that affect mental well-being (Rodríguez-Ayllón et al., 2019), in order to promote it from school based on a theoretical model.

In summary, as previously described, the authors justify the development of the present study in the little literature on the matter, and the importance of physical activity in the mental well-being of children and adolescents.

Introduction-pag 2, line 85-86 “Therefore, the present research aims to contribute to the improvement in mental well-being in children and adolescents through the proposal of a theoretical model.

Eddolls, W. T., McNarry, M. A., Lester, L., Winn, C. O., Stratton, G., & Mackintosh, K. A. (2018). The association between physical activity, fitness and body mass index on mental well-being and quality of life in adolescents. Quality of Life Research, 27(9), 2313-2320.

Garn, A.C.; Morin, A.J.S.; White, R.L.; Owen, K.B.; Donley, W.; Lonsdale, C. Moderate-to-Vigorous Physical Activity as a Predictor of Changes in Physical Self-Concept in Adolescents. Heal. Psychol. 2019.

Rodriguez-Ayllon, M., Cadenas-Sánchez, C., Estévez-López, F., Muñoz, N. E., Mora-Gonzalez, J., Migueles, J. H., ... & Esteban-Cornejo, I. (2019). Role of physical activity and sedentary behavior in the mental health of preschoolers, children and adolescents: a systematic review and meta-analysis. Sports medicine, 49(9), 1383-1410.

Hale, G. E., Colquhoun, L., Lancastle, D., Lewis, N., & Tyson, P. J. (2021). Physical activity interventions for the mental health and well‐being of adolescents–a systematic review. Child and Adolescent Mental Health.

Reviewer: The theoretical background is underdeveloped. The authors should create a theoretical background section and provide in-depth discussions on each construct in terms of how the constructs have been understood and examined in previous studies. Also, it is informative to talk about the importance of physical activity in high school students.

Response: Thank you very much for your suggestion for improvement, the authors appreciate the opportunity to improve the work. In this sense, we have added the next theoretical background:

Pag 2-3, line 92-105: The present theoretical model

“The present proposed theoretical model has been based on scientific evidence for its elaboration. In this regard, the following model postulates the role of physical activity through self-concept and subjective happiness in the quality of life of children and adolescents. To justify the model, we have relied on the relationship and predictive values of some variables with others. Likewise, previous published theoretical models have served as justification in the establishment of order of the variables [20,44,50]. For all these reasons, the present model proposes that physical activity and body mass index would be related and predict self-concept, subjective happiness, and quality of life in the last stay. Along these lines, several investigations show the importance of physical activity in the well-being of children and adolescents [51–53]. However, the benefits of this are due both to the time spent practicing physical activity and its intensity [20-21]. Both types of way of considering physical activity influence the perception of self-concept, making it better or worse [29], which, according to previous studies, could have effects on the well-being and emotional state of children and adolescents [18,34,35] that in the last stay would predict the quality of life they perceive [49].”

Reviewer:  Hypotheses developments are not provided. Please create another section and provide theories and literature that can support the hypotheses. Please also discuss why physical activity and BMI do not directly influence QOL or subjective happiness. Similarly, why did self-concept not directly affect QOL?

Response: Dear reviewer, there are no previous studies that have been fully developed as the hypothesized model that we present in this manuscript. Likewise, the authors on page 3, line 109 formulate the following hypothesis based on the theoretical model proposed according to the previous studies:

“In this sense, it is thought that physical activity and body mass index will predict quality of life through the mediating factors of self-concept and subjective happiness.”

Continuos…On the other hand, in relation to the question posed regarding the direct relationship between physical activity and BMI with quality of life. The authors indicate through table 1. (Correlation analysis) the direct relationships between all variables and quality of life. In this sense, it is shown that BMI was not significantly related. However, the rest of the variables showed significant positive direct relationships.

Likewise, since BMI was not significantly related to quality of life, we think that at an early age being fit or slightly above the percentage of body mass is not related to quality of life because at an early age child are in a growth stage and may have a BMI above or below normal weight. In this sense, a possible explanation could be the fat but fit paradigm holds that cardiovascular risk can be counteracted with cardiorespiratory fitness. Thus, the Martínez-Vizcaíno et al., 2021 study under this paradigm showed that there are no differences in the quality of life with respect to the percentage of body mass.

Martínez-Vizcaíno, V., Garrido-Miguel, M., Redondo-Tébar, A., Notario-Pacheco, B., Rodríguez-Martín, B., & Sánchez-López, M. (2021). The “Fat but Fit” Paradigm from a Children's Health-Related Quality of Life Perspective. Childhood Obesity.

Reviewer: I’m afraid to say that the findings look obvious, and thus I don’t see much contribution of the paper to the literature and practice. I believe there are many articles (as the authors cited) found similar results, and I question whether this paper can make a meaningful contribution that other studies have not discovered. Please discuss how and why the paper can advance our understanding in the area.

Response: Dear reviewer, we appreciate your opinion and agree with it. However, from our point of view to say that the findings are obvious if we pay attention to the relationships between variables separately, however, if we value the theoretical model in a general way, it is the first study that establishes predictions with the total of variables. In this way, we contribute to the research gaps in which there are few studies where physical activity, BMI and self-concept have been related to variables of well-being in adolescents (moods, satisfaction with life, subjective well-being) [29, 30], and even fewer have related self-concept to quality of life [44]. Therefore, we believe that our model contributes something new to the literature, since its practical application could contribute to the improvement of physical and mental well-being in children and adolescents, working from the attributes of physical condition to the improvement of well-being. psychological and affecting in the last stay the quality of life they perceive.

Reviewer: The authors said MRLx2 = 187.099, p< .05, CFI = .90, TLI = .87, SRMR = .09, and RMSA = .10 were not acceptable. Based on what evidence did the authors make the decision? There is no definitive fit index, meaning that it is just a mere recommendation. I don’t see any issues with the results, and I believe the authors should proceed with the original model.

Response: We welcome your suggestion for improvement. However, the statistical recommendations for the Tucker Lewis index or non-normalized index of adjustment indicate that it should be close to 1. Recommending the validity of the model at 0.90 (Browne & Cudeck, 1993). Similarly, Bentler, (1992) recommends that the CFI value should be greater than 0.90, indicating that at least 90% of the covariance can be produced by the model. Therefore, we decided to accept the adjustment indices with the recommended modifications (Figure 2).

BROWNE, M.W. y CUDECK, R., (1993): Alternative ways of assessing model fit. In A. Bollen and J.S. Long Eds., Testing Structural Equation Models, Sage Publications. Thousand Oaks, C.A.

Bentler, P. M. (1992). On the fit of models to covariances and methodology to the Bulletin.. Psychological bulletin, 112(3), 400.

Mangin, J. P. L. (2003). Modelización y análisis con ecuaciones estructurales. In Análisis multivariable para las ciencias sociales (pp. 767-814).

Reviewer: “the predictive role of perceived competence in self-esteem [55], physical condition on quality of life and finally the importance of BMI on appearance [56].” This is not good at all. Such a decision has to be based on a priori theory, not second through. I don’t understand how sub-constructs of self-concept can influence each other. Why did only BMI was expected to influence physical appearance? This is ill-theorized and not acceptable. Also, 5 sub-constructs of self-concept were conceptualized as formative (allows towards self-concept). Please provide rationales on the conceptualization.

Response: We appreciate your suggestion for improvement and consider your concern regarding the modifications required to build a model with acceptable fit rates. In this sense, we must point out that the proposed modifications have been made based on previous scientific literature in order to justify the theoretical relationships.

A continuación, procederemos a argumentar el motivo de estos ajustes: Siguiendo el orden del modelo, cuando se ejecutó el modelo hipotetizado en el programa estadístico, nos surgieron unos índices de ajustes que mejoraban la calidad de este. En este sentido, uno de los ajustes tenía que ver en el valor predictivo que posee la autoestima con la competencia percibida para la formación de la variable latente autoconcepto. A este respecto, contextualizándonos en la teoría referente a este constructo, el autoconcepto se considera como una variable multidimensional formada por distintos dominios. En este sentido, la propuesta de Fox y Corbin (1989) señala que el dominio más importante del autoconcepto es la autoestima, que es el juicio que una persona realiza a si mismo en como de satisfactorio es el desempeño de esta persona en áreas específicas y generales según su propio sistema de valores (Coopersmith, 1967; Piers, 1969). Seguidamente, se sitúan los dominios referentes a la competencia percibida, el fitness, la fuerza percibida, y la apariencia (Fox & Corbin, 1989). Así pues, la autoestima puede aumentar o disminuir según la competencia percibida, (Cole et al., 2001). Harter, añade que la competencia percibida es la medida en la que uno se juzga a si mismo como capaz en un área determinada de la vida, por ello mayor es la probabilidad de que presente una buena autoestima (Jekauc et al., 2019). Así pues, y citando el modelo de Harter justificamos el rol de la competencia percibida sobre la autoestima. Seguidamente, apoyándonos en la práctica de la actividad física, la cual conlleva una mejora en la condición física está se asocia directa e indirectamente con la calidad de vida y el Bienestar mental en adolescentes (Eddolls et al., 2018). Finalmente, para justificar el por qué el BMI influye en la apariencia, son muchos los estudios que aluden la relación que hay entre el BMI y la frustración satisfacción de la apariencia (Lin et al., 2018)[56]

Fox, K. R., and Corbin, C. B. (1989). The physical self-perception profile: development and preliminary validation. J. Sport Exerc. Psychol. 11, 408–433. doi: 10.1123/jsep.11.4.408

Coopersmith, S. (1967) The Antecedents of Self-Esteem. San Francisco, CA: W.H. Freeman.

Cole, D. A., Maxwell, S. E., Martin, J. M., Peeke, L. G., Seroczynski, A. D., Tram, J. M., Hoffman, K. B., Ruiz, M. D., Jacquez, F., & Maschman, T. (2001). The development of multiple domains of child and adolescent self-concept: a cohort sequential longitudinal design. Child Development, 72(6), 1723–1746.

Jekauc, D., Mnich, C., Niessner, C., Wunsch, K., Nigg, C. R., Krell-Roesch, J., & Woll, A. (2019). Testing the Weiss-Harter-Model: Physical Activity, Self-Esteem, Enjoyment, and Social Support in Children and Adolescents. Frontiers in psychology, 10, 2568.

Lin, Y. C., Latner, J. D., Fung, X. C., & Lin, C. Y. (2018). Poor health and experiences of being bullied in adolescents: Self‐perceived overweight and frustration with appearance matter. Obesity, 26(2), 397-404.

Altıntaş, A., Aşçı, F. H., Kin-İşler, A., Güven-Karahan, B., Kelecek, S., Özkan, A., ... & Kara, F. M. (2014). The role of physical activity, body mass index and maturity status in body-related perceptions and self-esteem of adolescents. Annals of human biology, 41(5), 395-402.

Therefore, the following explanation has been added in the results section, justifying the adjustments: Pag 5, line 209-216: “In this sense, based on the proposals of the theoretical model presented by Fox and Corbin (1989), which establish self-concept as a multidimensional variable formed by different domains, whose most important domain is self-esteem followed by perceived competence, fitness, perceived strength, and appearance. In this sense, perceived competence is the extent to which one judges oneself as capable in a certain area of life, which is why the greater the probability that they will present good self-esteem (Jekauc et al., 2019). Consecutively, it was adjusted according to the physical condition on quality of life in passing to its role on mental well-being (Eddolls et al., 2018).”

Reviewer: No practical implications were provided. Please discuss how the findings can help school and governmental policies.

Response: Thank you very much for your suggestion for improvement. In this regard, we have expanded the lines for practical applications in the concluding section.

Pag 7, line 323:327 “As practical implications, this theoretical model highlights the importance of performing physical activity with an adequate frequency and moderate vigorous intensity from an early age, since this fact contributes to an adequate mental development, improving the quality of life and avoiding that in later stages they develop different types of disorders in relation to their perception of self-concept. In the same way, the present theoretical model can contribute to promoting new methodologies when teaching classes in different subjects, incorporating innovative methodologies that favor the acquisition of knowledge through the practice of physical activity. In this way, we would contribute to the increase in daily minutes in the realization of physical activity and we would favor the mental well-being derived from the realization of physical activity.”

Reviewer 4 Report

In this manuscript, the authors studied how physical activity predicts the quality of life in adolescents. They restructured the equation model, and concluded that physical activity could improve self-concept and happiness, thus affect quality of life. They further showed that this model is acceptable for males, but not for females. Overall, this manuscript is interesting.

-Please discuss the potential reasons why this model presents a better fit for males than females. Is there any other factor involved? Such as types of physical activity?

Author Response

#Reviewer 4

Reviewer: In this manuscript, the authors studied how physical activity predicts the quality of life in adolescents. They restructured the equation model, and concluded that physical activity could improve self-concept and happiness, thus affect quality of life. They further showed that this model is acceptable for males, but not for females. Overall, this manuscript is interesting.

-Please discuss the potential reasons why this model presents a better fit for males than females. Is there any other factor involved? Such as types of physical activity?

Responds: Dear reviewer, we appreciate the comments received from you and the suggestion for improvement, as it adds more rigor to this investigation. In this sense, we find your proposal to discuss the differences of fit in the model according to gender very interesting. Therefore, we have prepared a new paragraph in the discussion to respond to your suggestion.

Pag 7, line 286-301 “Finally, it is important to note that this model presented fit indices that are more acceptable for boys than for girls. In this regard, there are several causes that could explain these findings, such as adolescence, which is characterized by a decrease in levels of physical activity, especially among girls (Corder et al., 2019; Sousa-Sa et al., 2020). This fact could be explained through the amount of daily physical activity performed by boys or girls (Dalton et al., 2011), or the different types of physical activity practiced according to gender, which is associated with gender stereotypes, causing rejection of certain activities due to perceive them as masculine (Corr et al., 2018). In this regard, the girls tend to prefer activities related to body shape and health with a more aesthetic orientation, preferring individual sports, while boys tend to opt for activities focused on improving fitness or physical performance, choosing team sports in which strength and competitiveness predominate (Chacon-Cuberos et al., 2016; Peral-Suarez et al., 2020 In the same way, another possible explanation can be found in the perception of weight and body mass index, which could affect girls more significantly than boys, causing damage to self-esteem according to their perceived appearance, and affecting their well-being (Guerrero et al., 2020). In this regard, the study by Vaquero-Solís et al. (2021) based on the theory of self-objectification shows that physical activity in girls is more associated with appearance and self-esteem.”

Corder, K., Winpenny, E., Love, R., Brown, H. E., White, M., & Van Sluijs, E. (2019). Change in physical activity from adolescence to early adulthood: a systematic review and meta-analysis of longitudinal cohort studies. British journal of sports medicine, 53(8), 496-503.

Peral-Suárez, Á., Cuadrado-Soto, E., Perea, J. M., Navia, B., López-Sobaler, A. M., & Ortega, R. M. (2020). Physical activity practice and sports preferences in a group of Spanish schoolchildren depending on sex and parental care: a gender perspective. BMC pediatrics, 20(1), 1-10.

Vaquero-Solís, M., Tapia-Serrano, M. A., Moreno-Díaz, M. I., Cerro-Herrero, D., & Sánchez-Miguel, P. A. (2021). Análisis exploratorio de la actividad física en la auto-objetificación e insatisfacción corporal de jóvenes adolescentes (Exploratory analysis of physical activity in self-objectification and body image of adolescents). Cultura, Ciencia y Deporte, 16(48), 199-206.

Guerrero, M. F., Molina, S. F., & Ramírez, M. S. (2020). Physical self-concept in terms of sociodemographic variables and their relationship with physical activity. Cultura, Ciencia y Deporte, 15(44), 189-199. https://doi.org/10.12800/ccd.v15i44.1461

Round 2

Reviewer 3 Report

I appreciate the authors trying to responding my comments.

But there are many parts where my previous comments were not addressed. I reiterated below.

- The previous comment about the lack of motivation for the study has been addressed. Again, state WHY it’s important to examine this study, instead of explaining it to me in the letter.

- If there are no previous studies and theories that can support hypotheses, the authors should not use the term “predict.” Please change the hypotheses to research questions.

- For my previous comments on the lack of contributions, instead of explaining it to me, incorporate it in the manuscript and strengthen the part.

- For your response “We welcome your suggestion for improvement. However, the statistical recommendations for the Tucker Lewis index or non-normalized index of adjustment indicate that it should be close to 1. Recommending the validity of the model at 0.90 (Browne & Cudeck, 1993). Similarly, Bentler, (1992) recommends that the CFI value should be greater than 0.90, indicating that at least 90% of the covariance can be produced by the model. Therefore, we decided to accept the adjustment indices with the recommended modifications (Figure 2).” I don’t think that is a valid one. CFI is .90. TLI is .87. And most all, the authors did not address the comments in the manuscript. I still strongly believe this is ill-practice, and the authors should use the ORIGINAL model.

- Practical implications should be further strengthened.

Author Response

Responds to reviewer

Dear reviewer, we again appreciate the opportunity to improve the quality of the manuscript and discuss the results obtained. In this regard, we would like to point out that the authors have tried to provide answers to each one of your suggestions for improvement. In this sense, we have always relied on scientific evidence as the basis for the elaboration of our answers. Finally, we hope that our contributions be considered and appreciated.

I appreciate the authors trying to responding my comments.

But there are many parts where my previous comments were not addressed. I reiterated below.

Reviewer. - The previous comment about the lack of motivation for the study has been addressed. Again, state WHY it’s important to examine this study, instead of explaining it to me in the letter.

Authors. We appreciate your contributions. In this sense, we try to respond to your suggestion by following your recommendation and adding the reasons to the text.

Therefore, on page 2, fourth paragraph, lines 84 to 93, we highlight the reasons for conducting this study. In this sense, throughout the paragraph, there are 3 fundamental ideas that explain the motivation of the study: 1) the few investigations that have jointly addressed the variables of interest of the study, 2) the impact of the different types of physical activity (intensity and duration) on self-concept and mental well-being, and 3) the small number of investigations that associate self-concept with quality of life. Therefore, the present research aims to contribute to the improvement of mental well-being in children and adolescents through the proposal of a theoretical model based on the importance of physical activity as a predictor of health-related quality of life (Haegele et al., 2017) considering constructs that affect the development of psychological well-being of adolescents (Corder et al., 2020) such as subjective happiness and quality of life.

References:

Haegele, J.A.; Famelia, R.; Lee, J. Health-related quality of life, physical activity, and sedentary behavior of adults with visual impairments. Disabil. Rehabil. 2017, 39, 2269–2276, doi:10.1080/09638288.2016.1225825.

Corder, K.; Werneck, A.O.; Jong, S.T.; Hoare, E.; Brown, H.E.; Foubister, C.; Wilkinson, P.O.; van Sluijs, E.M. Pathways to Increasing Adolescent Physical Activity and Wellbeing: A Mediation  Analysis of Intervention Components Designed Using a Participatory Approach. Int. J. Environ. Res. Public Health 2020, 17, doi:10.3390/ijerph17020390.

Reviewer. If there are no previous studies and theories that can support hypotheses, the authors should not use the term “predict.” Please change the hypotheses to research questions.

Authors: We would like to thank your comment. We agree with reviewer, therefore, we have changed the term predictive for the explanatory term in the objectives. Page 3, line 114.

In addition, we have reformulated the paragraph of the objectives following the scientific method from which we ask ourselves a question, and from it the object of study and the hypotheses are derived.

Pag 3, line 112-118 “Therefore, the following research question arises: can physical activity and body mass index explain the quality of life of adolescents through the mediating factors of self-concept and subjective happiness? In this sense, the objective of the research is to analyze the explanatory value of physical activity on quality of life through self-concept and subjective happiness. In addition, it is proposed to determine the validity of the theoretical model for both sexes. In this regard, it is thought that physical activity and body mass index will predict quality of life through the mediating factors of self-concept and subjective happiness.”

Reviewer. - For my previous comments on the lack of contributions, instead of explaining it to me, incorporate it in the manuscript and strengthen the part.

Authors: Thank you very much for your suggestion for improvement. In this regard, the authors have expanded the strengths and contributions of the manuscript.

Page 8, line 311-325. "Regarding the strengths, this study has contributed to increase the existing literature through its contributions on two research topics. On the one hand, it contributes to increasing the scientific knowledge that relates physical activity, body mass index and self-concept with variables of well-being in the adolescent population (moods, satisfaction with life, subjective well-being) [29, 30]. On the other hand, it contributes to the small number of studies that relate self-concept with quality of life [44]. In addition, a predictive model is provided in a mixed adolescent population, as a similar model had previously been tested in a population of adult women [44]. Similarly, the indirect effects of physical activity and BMI with quality and life are shown through the partial mediating value of self-concept and mood. Finally, it contributes to provide an explanation of well-being in the male and female gender derived from the types of physical activity and the importance of appearance and self-esteem in one gender and another. Thus, it is emphasized that our model contributes something new to the literature, since its practical application could contribute to the improvement of physical and mental well-being in children and adolescents, working from the attributes of physical condition to the improvement of psychological well-being, and affecting in the last stay the quality of life they perceive.”

Reviewer. For your response “We welcome your suggestion for improvement. However, the statistical recommendations for the Tucker Lewis index or non-normalized index of adjustment indicate that it should be close to 1. Recommending the validity of the model at 0.90 (Browne & Cudeck, 1993). Similarly, Bentler, (1992) recommends that the CFI value should be greater than 0.90, indicating that at least 90% of the covariance can be produced by the model. Therefore, we decided to accept the adjustment indices with the recommended modifications (Figure 2).” I don’t think that is a valid one. CFI is .90. TLI is .87. And most all, the authors did not address the comments in the manuscript. I still strongly believe this is ill-practice, and the authors should use the ORIGINAL model.

Authors: Many thanks for your suggestion. We would like to suggest you reconsidering your opinion regarding the fit index. In this sense, and with due respect, most of the articles that expose structural equation models speak of fit index for the CFI and TLI (Tucker Lewis Index), also known as Non-normed Fit Index (NNFI) of coefficients. superiors. to .90 and in some cases from .95. For this reason, it was decided to make modifications to the model, the objective of which was to improve the fit indices. In this sense, the modifications were not made from our conviction, as we relied on scientific evidence to see if it supported these modifications. And for this reason, we justify the adjustments from line 217 to line 224. Also, I leave you again the articles that expose the recommendations of the fit index of the previous answer, and new references that support said indices of recommended adjustments greater than. 90 the statistical recommendations for the Tucker Lewis index or non-normalized index of adjustment indicate that it should be close to 1. Recommending the validity of the model at 0.90 (Browne & Cudeck, 1993). Similarly, Bentler, (1992) recommends that the CFI value should be greater than 0.90, indicating that at least 90% of the covariance can be produced by the model. Similarly, Hu and Bentler (1999), and Kline (2004) are much severer accepting CFI and TLI coefficients above .95.  For these reasons, I ask that you reconsider your judgment of acceptable fit index.

Hu, L., & Bentler, P. M. (1999). Cutoff criteria for fit indexes in covariance structure analysis: Conventional criteria versus new alternatives. Structural Equation Modeling, 6, 1–55. https://doi.org/10. 1080/10705519909540118

Kline RB.Principles and Practices of Structural Equation Modeling.2nd ed. New York, NY:Guilford Publications Inc; 2005.

Reviewer. - Practical implications should be further strengthened.

Authors: Thank you so much for the contribution, we appreciate your suggestion for improvement. In this regard, the practical implications have been expanded from lines 331 to 343.

“As practical implications, this theoretical model highlights the importance of performing physical activity with an adequate frequency and moderate vigorous intensity from an early age, since this fact contributes to an adequate mental development, improving the quality of life and avoiding that in stages subsequent development. different types of disorders in relation to their perception of self-concept. Similarly, this theoretical model can contribute to promoting new methodologies when teaching classes in different subjects, incorporating innovative methodologies that favor the acquisition of knowledge through the practice of physical activity. In this way, we would contribute to the increase in daily minutes in physical activity and promote mental well-being derived from physical activity. In the same way, it would be important to support ourselves in the interests of a similar physical activity for the promotion of sports practice within the educational context, since this would avoid the rejection of the most sedentary people and with a worse perception of their self-concept, for the carrying out physical activity, from an integral approach to education for the promotion of well-being with others and with oneself.”
